# Parameter Optimization of Hole-Slot-Type Magnetron for Controlling Resonant Frequency of Linear Accelerator 6 MeV by Reverse Engineering Technique

**Nattawat Yachum** [1,2] , **Somjai Chunjarean** [1] , **Nilaped Russamee** [1] **and Jiraphon Srisertpol** [2,*]

1  Synchrotron Light Research Institute, Nakhon Ratchasima 30000, Thailand; nattawat@slri.or.th (N.Y.); somjai@slri.or.th (S.C.); nilaped@slri.or.th (N.R.)
2  School of Mechanical Engineering, Suranaree University of Technology, Nakhon Ratchasima 30000, Thailand
*  Correspondence: jiraphon@sut.ac.th

**Abstract:** This paper presents the parameter optimization of a twelve-hole-slot-type magnetron based on a reverse engineering technique to improve a 6-MeV linear accelerator (LINAC) operation for fruit sterilization. The magnetron structural dimensions are measured by a coordinate measuring machine (CMM) that has tolerance on a dimension of 0.5 μm to analyze the resonant frequency with a desired operating point of the magnetron in the dominant mode. There are two methods of analysis using a proper parameter for the magnetron operation. The first method is mathematical model analysis of an equivalent resonant parallel circuit. The other is 3D-model drawing of the magnetron based on particle-in-cell (PIC) using computer simulation technology microwave studio (CST). The results are demonstrated by the position of the resonant frequency of each mode of operation, and the radius and tuner slot distance of a cavity within the structures of the magnetron cause a resonant frequency change. The suitable parameters of the voltage and magnetic field supply are desired to control a resonant frequency at 2.9982 GHz by using the Takagi–Sugeno fuzzy logic control (FLC) algorithm to control a resonant frequency at 2.9982 GHz. The results of the FLC algorithm application show that the LINAC can produce X-rays with a constant dose rate for an hour with a disturbance in the range of 38 to 42 °C temperature and $1 \times 10^{-9}$ to $5 \times 10^{-8}$ torr vacuum pressure.

**Keywords:** equivalent circuit; hole-slot-type magnetron; resonant frequency; particle-in-cell; linear accelerator; fuzzy logic control

## 1. Introduction

Agriculture productivities used to be important drivers of Thailand's economic growth. There are many exported agricultural products, including rice and a variety of fruits and vegetables. Recently, many countries have imposed strict regulations on imported fruits as a result of pathogenic microorganisms. To ensure the safety of fresh fruits and vegetables and prevent the introduction of unwanted diseases, irradiation with accelerator machines is widely used to improve the physical, chemical or biological properties of the products. Synchrotron Light Research Institute (SLRI), Thailand, has researched, developed and designed the prototype of a Linear Accelerator (LINAC) for X-ray sterilization with an electron energy of 6 MeV. The LINAC system produces the X-ray by accelerating the 6-MeV electron beam to hit a target with high Z. The target used to convert the electron beam to bremsstrahlung is tungsten. The acceleration of the electron beam requires three main components, consisting of an electron gun, an accelerator structure and a radio frequency generator or magnetron that can oscillate and produce an electromagnetic field as a standing wave within each cavity of the accelerator tubes.

A magnetron is a high-power oscillator generating an electromagnetic field and used in a wide range of applications including military radar, aerospace, navigation and medical use [1–6]. Among the various high-power oscillators, a magnetron can be manufactured

with compact size, high efficiency, light weight and simple structure. E2V, located in Chelmsford, has been manufacturing magnetrons for civil marine applications since 1947 based on an S-band design, and they have developed an X-band magnetron. In the 1950s, a magnetron adapted for use in photon beam radiotherapy was developed which requires a pulsed RF source to accelerate electrons in the cavities of LINAC [7]. The different physical structures of the anode block in a magnetron are sector, hole-slot, rising sun, ellipse and vane-type [8–10]. The applied voltage between the anode and cathode in the magnetron-producing DC field and the magnetic field is applied perpendicular to the electric field in the interaction space and resonant cavities in the anode block. If the values of the electric and magnetic field are suitable, a resonance condition occurs and the desired microwave frequency is generated by the cavities. The dimensions and shape of the cavities and the interaction space between the anode and cathode determine the operation frequency of magnetrons [10]. There are several methods to determine a magnetron's resonant frequency, such as theoretical equivalent circuit of geometry analysis, field theory, computer simulation and experimental measurement. Each method has different advantages. The theoretical equivalent circuit method can obtain a resonant frequency formula as a function of the structure dimensions of a magnetron according to the relation between resonant frequency and structure dimensions. Thereby, it can analyze the effect of the structures on the resonant frequency of the magnetron [11]. The field theory is very sophisticated and it requires advanced mathematics to solve the electromagnetic equation. Computer simulation is usually used for design and optimization, but it requires long periods of time. Particle-in-cell (PIC) simulation is the electromagnetic solver in CST Particle Studio (CST-PS), which is widely used for magnetron design. It is based on finite integration for the simulation of electromagnetic fields in the desired model and is usually used to estimate the fundamental mode frequency of a relativistic magnetron and tune the resonant frequency. The last method is the frequency measurement, which is not suitable for magnetron design, but it is well suited to the practical application of Hole-Slot-Type magnetrons.

In this paper, we focus on the effects of disturbances such as temperature and vacuum pressure on the twelve-hole-slot-type magnetron which is used for the 6-MeV LINAC. The disturbance affects the resonant frequency change and mismatch impedances between the frequency source and output load (accelerator tube) during the LINAC operation. These effects lead to electron acceleration, which would not gain sufficient energy to travel through the end of the LINAC. Thus, the X-ray quantity used for the irradiation is either insufficient or too high to sterilize products. To achieve an appropriate frequency and proper characteristics regarding the modes of the magnetron during the LINAC operation, the measurement of the structural dimensions and shape of the magnetron and simulation method are applied to analyze the resonant frequency of the magnetron. This paper is organized in three main parts. First, the dimensional parameters and shape of the magnetron are measured by a coordinate measuring machine (CMM) with tolerance of 0.5 µm. To analyze these parameters, the resonant frequency at a suitable operating point in the dominant mode is achieved by using equivalent resonant circuit analysis and 3D-drawing analysis, which are based on PIC simulation in the CST Particle Studio program. The PIC simulation allows the analysis to vary and adjust the virtual parameters and structural dimensions of the magnetron and observe dynamic changes in these parameters during the simulation. The last is the design of the LINAC control system to obtain stable operation of the magnetron in the dominant mode by controlling certain parameters, such as input anode voltage V and the magnetic field B. The control system is based on a fuzzy logic control (FLC) algorithm with the Takagi–Sugeno fuzzy rule used to represent local parameters in relation to a nonlinear system. The approach of the control system is to adequately match the LINAC's resonant frequency or any load with a magnetron frequency of 2.9982 GHz to maximize the power used for electric field gradient generation in the LINAC. This serves to maintain the emission of the high-energy X-ray beam at the desired absorbed dose rate during the irradiation of agricultural products. Here, the RF

source or the magnetron frequency is tracked and adjusted according to the time variation of the resonant frequency by controlling the shaft of a stepping motor that is coupled with the internal structure of the magnetron. Under these experimental conditions, both temperature and vacuum pressure are maintained in the range of 38 to 42 Celsius and $1 \times 10^{-9}$ to $5 \times 10^{-8}$ torr, respectively, to control for environmental disturbance.

The structure of the paper is as follows: Section 2 describes the research methodology, consisting of three subsections. Firstly, an outline of the magnetron structure analysis is presented, which gives details of the measurement of the structural magnetron and parameter optimization. The second section presents the analysis of the magnetron frequency with the equivalent circuit and PIC simulation method. The third section presents the fuzzy logic control algorithm and system for the sterilization of agricultural products with an X-ray beam. Finally, the experimental results and the conclusions are presented in Sections 3 and 4, respectively.

## 2. Research Methodology

The frequency control of a twelve-hole-slot-type magnetron for a 6-MeV LINAC applied for sterilizing agricultural products is studied by using the reverse engineering method. As a result, optimized parameters of the structure dimension and shape of the twelve-hole-slot-type magnetron are achieved in dominant-mode operation. A magnetron used for a medical linear accelerator is used in this research [12]. The procedures of the research methodology are described in the following section.

### 2.1. Magnetron Structure Analysis

One of the most efficient sources of high-power microwave is a relativistic magnetron. A relativistic magnetron has the structure of a diode which mainly consists of the cathode and anode polarities confining electrons in between. The anode of the magnetron is usually manufactured by a cylindrical copper block configuration. The cathode and filament are installed in the central area of the anode block and supported by the filament leads. The filament leads are rigid in order to maintain the position of the cathode and filament. The anode is indirectly supplied with DC voltage to heat up the filament. This leads to electrons being emitted from the cathode surface, which is constructed from a high-emission material around the area of the cavity. In general, the cavity magnetron has been designed to possess eight or twenty cavities distributed around the cathode. It is called the resonant cavity, where each cavity forms a parallel resonant circuit, as shown in Figure 1.

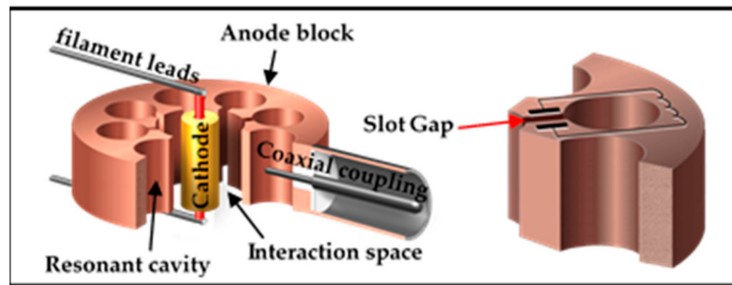

**Figure 1.** The structures and resonant cavity of the magnetron.

Figure 1 shows the magnetron configuration, which consists of several resonant cavities, a space between the anode block and cathode called the interaction space and a slot gap which runs from each cavity into the central part of the cathode. The overall resonant cavity is enclosed by metal walls equivalent to inductance, while the slot gap represents the capacitive portion of the equivalent parallel resonant circuit. Thus, to simplify the study, the lumped equivalent circuit of a resonator is used to represent the cavity with the parallel combination of an inductor and a capacitor, while the dimension and shape of the cavity represent the resonant frequency. Electrons emitted from indirect heat in the cathode travel radially towards the anode if the magnetic field produced by a permanent magnet is zero.

If the magnetic field strength is slightly increased, the magnetic force will bend the electron path as a circular path to the outer ring, which is the anode. However, the magnetic field's strength, at some point, prevents the electrons from reaching the anode and returns them to the cathode, causing overheating in the cathode. Thus, this results in zero current in the anode. The circular path of the electron will return to the cathode without reaching the anode's surface when the magnetic field is less than the critical magnetic field ($B_C$). Because the electrons' trajectory changes during travel to the anode, they will lose their energy by radiating electromagnetic energy. This radiated electromagnetic wave is the natural resonant frequency of the cavity. In addition, these electrons create a charge on the anode's surface, leading them to radiate an electromagnetic wave at the resonant frequency. As a result, the cavity is excited and then it induces electromagnetic energy oscillation in the adjacent cavity with a phase of 180 degrees. The resonant frequency operate like a slow-wave structure or multi-cavity traveling-wave magnetron. In the cathode part, the transfer of the electron energy that is emitted by filaments moves to the interaction space, where the electric and magnetic fields interact perpendicularly together and cause a force to hit the electron, resulting in a loss of the energy of the electron partly to the area of the cavities. The output of the resonant frequency is coupled by coaxial coupling to the target load.

### 2.1.1. Geometric Dimensioning and Tolerancing

Geometric dimensioning and tolerancing (GD and T) is a technique used to plan industrial production and communication between a producer and designer, including dimensional inspection of the product to ensure good quality and high precision according to the designer as required. The theoretical analysis of the twelve-hole-slot-type magnetron operation is conducted using precise data on the magnetron's structure. Then, using this information as the initial parameters in the simulation and analysis in the CST Particle Studio program to achieve the dominant mode of the magnetron operation. To measure the dimensions of the magnetron structure with high accuracy, a coordinate measuring machine or CMM is used to mechanically probe the object in 3D-coordinate geometry; then, the reference plane of this object is constructed in order to obtain data on all dimensions with 0.5-μm-length accuracy from GD and T analysis.

Figure 2 shows the structural dimensions of the magnetron, measured by using the probe tip of the CMM machine to touch the target points of each component as required, such as a circular or rectangular geometry. To evaluate the magnetron dimension more precisely and with reliable measurement, more measurement points are required. Typically, the probe tip is used to touch the magnetron structure point by point for many points of measurement. In addition, the measurement of the dimensions is done at least 3 times, resulting in 0.5-μm tolerance. For all collected measured points, x, y and z coordinate planes are constructed using the computer-numerical-control in the CMM. The structural dimensions of the magnetron are obtained and are used to calculate and analyze suitable parameters which are required for the magnetron's operation in dominant mode.

### 2.1.2. Interaction between Electrons and Fields

An electron's motion in the interaction space depends on the orientation of the electric and magnetic fields. The magnetic field is perpendicular to the direction of electron motion, resulting in an electron trajectory in the form of a spiral path and in the direction of the electric field. Thus, both electric and magnetic fields are employed in the magnetron's operation and these fields exist in perpendicular directions so that they intersect. As a result, the magnetron is a crossed-field device. The crossed field is operated depending on the direction of each field, both of which exist at right angles to each other. In the crossed-field tube, electrons emitted from the cathode are accelerated by the electric field and gain velocity; however, the greater their velocity, the more their path is bent by the magnetic field. The electron charge is influenced by the electric and magnetic fields, which can be described as acting on the electron because of the presence of both fields due to the

Lorentz force law and the equation of motion for an electron in cylindrical coordinated form, which are expressed as follows [13–16]:

$$\frac{d^2r}{dt^2} - r\left(\frac{d\phi}{dt}\right)^2 = -\frac{e}{m}\left(E_r + B_z r\frac{d\phi}{dt} - B_\phi\frac{dz}{dt}\right),\tag{1}$$

$$r\frac{d^2\phi}{dt^2} = -\frac{e}{m}\left(E_\phi + B_r\frac{dz}{dt} - B_z\frac{dr}{dt}\right),\tag{2}$$

$$\frac{d^2z}{dt^2} = -\frac{e}{m}\left(E_z + B_\phi\frac{dr}{dt} - B_r r\frac{d\phi}{dt}\right),\tag{3}$$

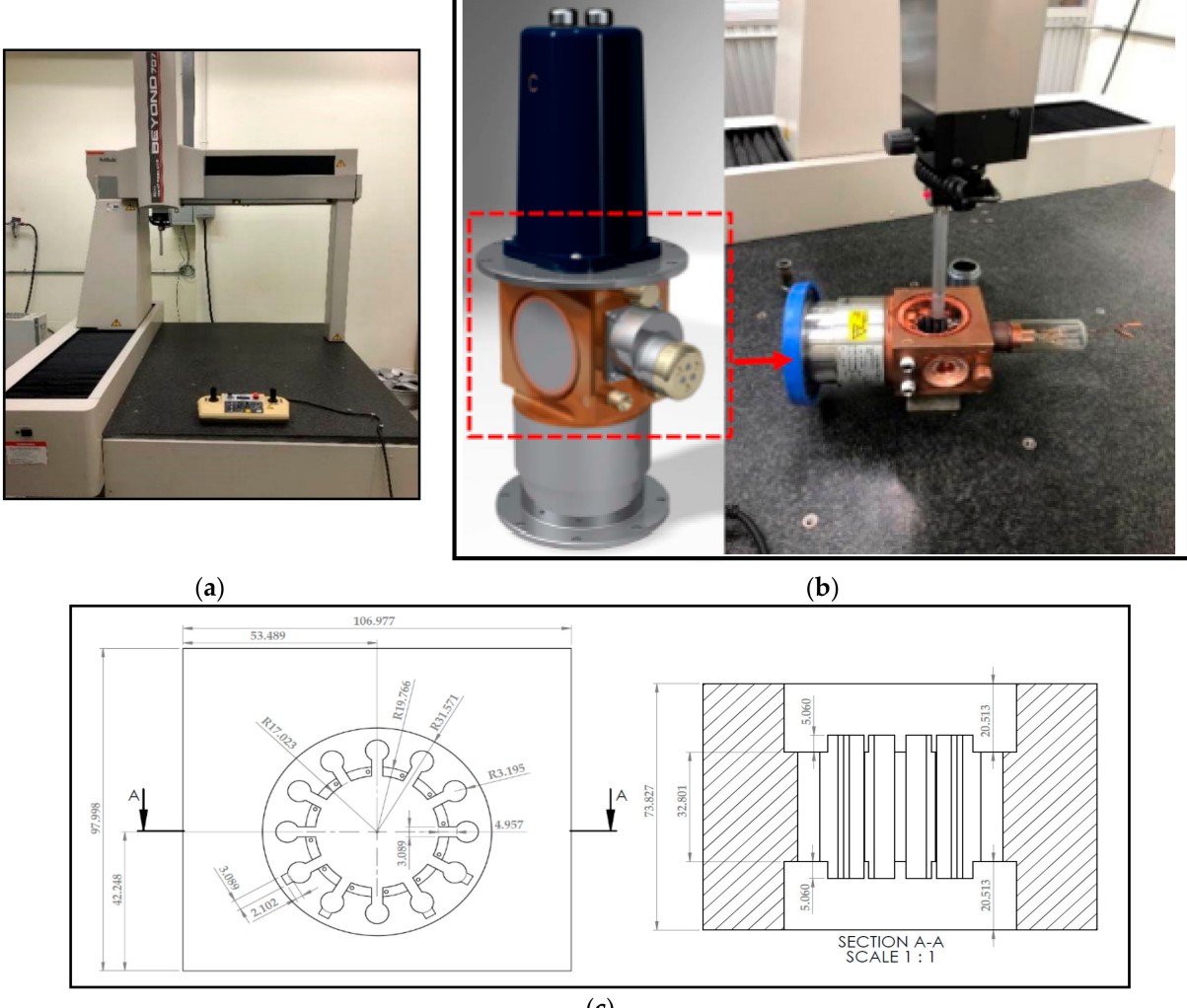

**(a)**　　　　　　　　　　　　　　　　　　　　　**(b)**

**(c)**

**Figure 2.** Structural dimensions measured by using coordinate measuring machine. (**a**) CMM Mitutoyo Beyond 707; (**b**) Internal structure measurement; (**c**) 3D diagram.

### 2.1.3. Hull Cut-off and Hartree Conditions

The electron motion in the magnetic field within the cylindrical radius of the cathode ($R_C$) and the radius of the anode ($R_A$) are excited by applying a high voltage ($V_{0C}$) and a permanent magnetic field ($B_{0C}$), with the magnetic field directed towards the page. The appropriate voltage and magnetic field are applied in order to cause the electron to be emitted from the cathode to the surface of the anode and then backwards to the interaction space area, as shown in Figure 3.

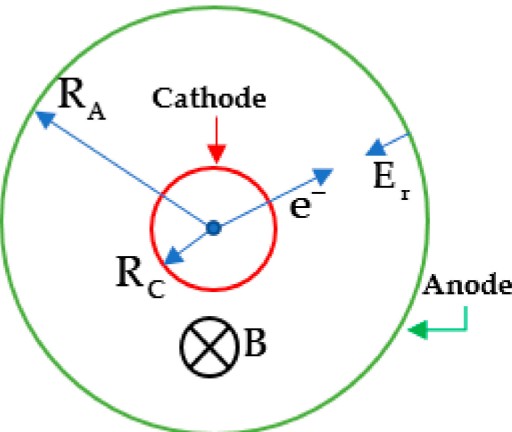

**Figure 3.** An electron's motion in an electromagnetic field.

The boundary of the magnetron operation is ensured by using the Hull cut-off and Hartree conditions; these represent the range of the oscillation effect of the magnetron. The Hull cut-off condition is described as the maximum range of the magnetron's operation that can occur if an electron moves to the adjacent anode block with zero velocity in the radius of the anode and assuming that the electron motion moves towards the cathode, referring to the laws of energy conservation. The Hartree condition, which represents the minimal range of the magnetron's operation, determines the anode voltage and magnetic field that are necessary to obtain an anode current with a non-zero value. The electron's motion in the interaction space around the cathode is determined by the magnetic field, which adheres to the laws of energy conservation and the condition of electron motion in a cylindrical coordinate that moves around the radius of the cathode with the same radius. Therefore, the equations to express anode voltage using the Hull cut-off and Hartree conditions are expressed as follows:

$$V_{0C} = \frac{e}{8m}(B_{0C})^2 \left( \frac{R_A{}^2 - R_C{}^2}{R_A} \right)^2, \tag{4}$$

$$V_H = \frac{\omega_0 B}{2} \left( R_A{}^2 - R_C{}^2 \right) - \frac{m}{2e} R_A{}^2 \omega_0{}^2, \tag{5}$$

where $\omega_0 = \frac{\omega}{n}$, $\omega$ is resonant frequency, $n = \frac{N}{2}$, is mode operation, N is number of cavities, e = electron charge ($1.6 \times 10^{-19}$ coulombs), m = electron mass ($9.11 \times 10^{-31}$ kg).

*2.2. Theoretical Magnetron Analysis*

A relativistic magnetron with several resonant cavities involves the operation of both fields perpendicularly, including the electric and magnetic fields. The exciting mode of operation of the magnetron can be utilized in many modes and results in different resonant frequencies, but the operation of the magnetron must predominantly take place in one mode only in order to control the resonant cavity. The typical analysis of the positional resonant frequency depending on the proper parameters of the internal structures of the magnetron can be carried out using two methods, consisting of the equivalent circuit method analysis and the PIC simulation method of analysis based on the CST program.

2.2.1. The Analysis of Resonant Frequency by Equivalent Circuit Method

The resonant cavity system takes into account the passive electrical circuit that is used in the oscillator. The normally resonant circuit is divided into 2 parts that consist of the series and parallel resonant circuit. The type of coupling resonant cavity consists of a coaxial line and aperture. The magnetron is presented in Figure 4 using the structure of coupling to a coaxial line with the resonant cavity.

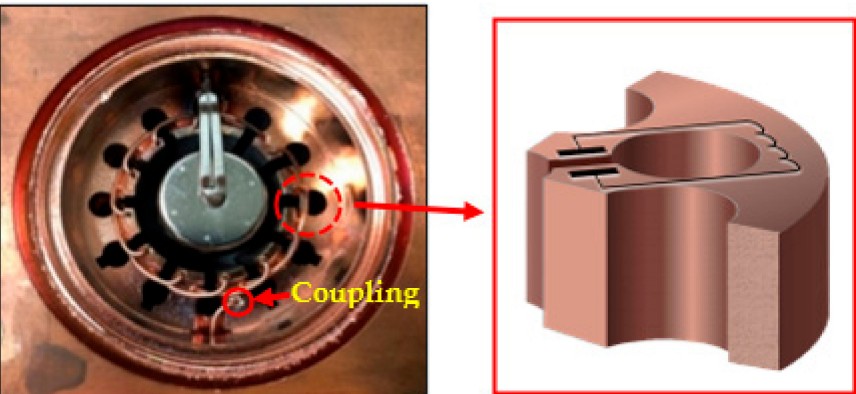

**Figure 4.** Internal structures of twelve-hole-slot-type magnetron and equivalent circuit of resonant cavity.

In Figure 4, the coupling type of the coaxial line is illustrated by inducing a current and voltage to transmit the coaxial line within matching impedance for the resonant circuit system. The energy of electromagnetic field coupling in each cavity of the resonant circuit is represented by a parallel circuit of capacitance, inductance and conductance; the equivalent circuit of the single-hole-slot magnetron resonant system is shown in Figure 5, where C, L and G are equivalent to capacitance, inductance and conductance, respectively. Thus, the resonant frequency of the equivalent parallel circuit of the magnetron is given by Equation (6). In Figure 5b, the consideration of an equivalent one-section network for the interaction space is described in terms of the network. It represents a simple magnetron operation that emits an electron from the cathode to interact with an electric field within the area of the interaction space. The equivalent network takes into account the capacitance ($C_{AK}$) between the anode segment and cathode. The equivalent resonant circuit of the magnetron is shown in Figure 5c [17–20].

$$\omega_0 = \frac{1}{\sqrt{LC}},\tag{6}$$

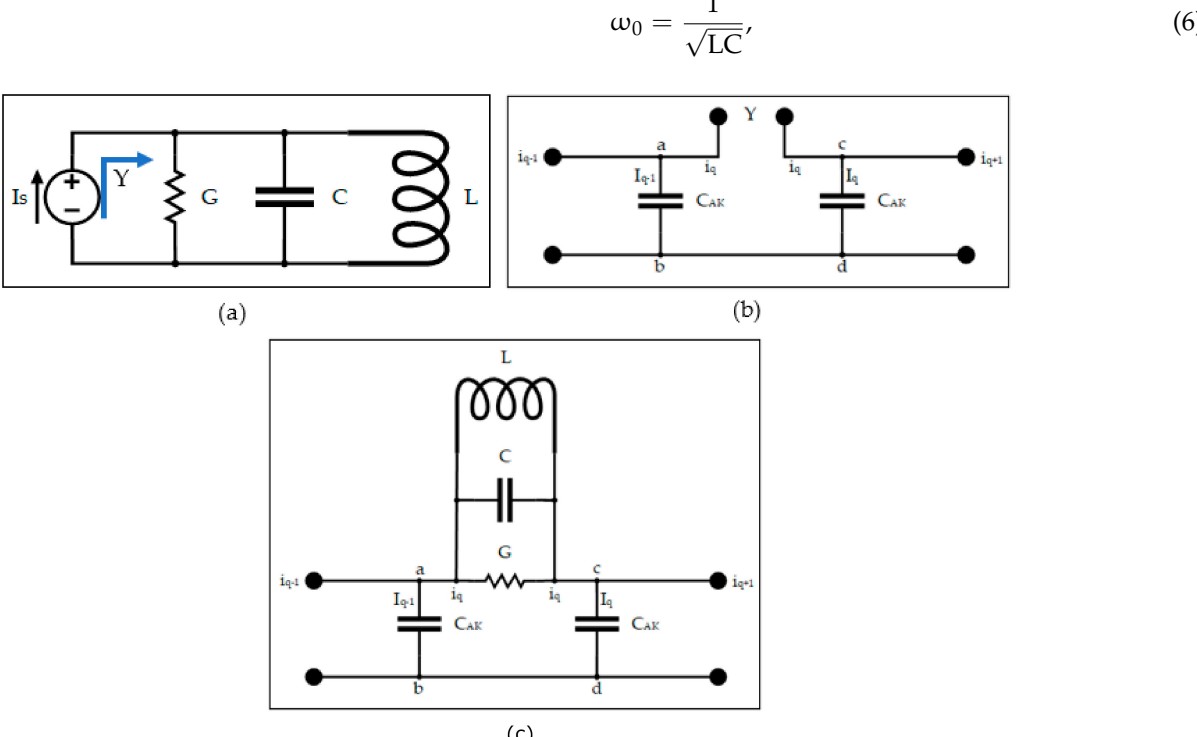

**Figure 5.** The equivalent resonant circuit network of the magnetron. (**a**) The equivalent circuit for the side resonators; (**b**) the equivalent circuit for the interaction space; (**c**) the equivalent circuit of the magnetron.

Analysis of the admittance values shown in Figure 5a,b when the resonance of the circuits occurs was conducted. The admittance values of the two-section configuration are equal. This means that both circuits can be rearranged and then the resonant frequency that occurs can be analyzed, as shown in Figure 5c. Moreover, a simple equation expressing the equivalent resonant circuit of the magnetron in each mode of operation is given as Equation (7).

$$\omega(\phi) = \frac{1}{\sqrt{(LC)\left(1 + \frac{C_{AK}}{2C(1 - \cos\phi)}\right)}}, \tag{7}$$

where $\phi = \frac{2\pi n}{N}$, $n = \frac{N}{2}$, is mode of operation, N is number of cavities.

Using Equation (7), the capacitance C, $C_{AK}$ and inductance L of the resonant system together with the interaction space shown in Figure 6 can be analyzed. The capacitance value is considered the electric distribution required to calculate the maximum electrical voltage change occurring on the adjacent anode block to the point at the voltage as zero or the upper area of the resonant cavity. This is expressed in Equations (8) and (9). For the inductance equation, it analyzes the area of the resonant cavity that is equivalent to the magnetic field in the conductor coil where the current flows in a closed loop. It can be described based on Ampere's law so that the density of the current flow in the slot area and the resonant cavity of the magnetron are obtained. As a result, the inductance is identified by using the laws of Faraday, which are expressed in Equation (10). Thus, the resonant frequency of each mode of the twelve-hole-slot-type magnetron is given in Equation (11) [12,17,21].

$$C = \frac{\varepsilon_0 H L_M}{W}, \tag{8}$$

$$C_{AK} = \frac{2\pi\varepsilon_0 H}{N\ln\left(\frac{R_A}{R_C}\right)} + \frac{\varepsilon_0 HW}{R_A\ln\left(\frac{R_A}{R_C}\right)}, \tag{9}$$

$$L = \frac{\mu_0 R_v}{24H} + \frac{\mu_0 W L_M}{2H} \tag{10}$$

$$f_r(n) = \frac{1}{2\pi\sqrt{\frac{\mu_0}{2H}\left(\frac{R_v}{12} + WL_M\right)\left(\frac{\varepsilon_0 L_M H}{W}\right)\left(1 + \frac{\frac{2\pi\varepsilon_0 H}{N\ln\left(\frac{R_A}{R_C}\right)} + \frac{\varepsilon_0 HW}{R_A\ln\left(\frac{R_A}{R_C}\right)}}{\left(\frac{2\varepsilon_0 H L_M}{W}\right)\left(1 - \cos\frac{2\pi n}{N}\right)}\right)}}, \tag{11}$$

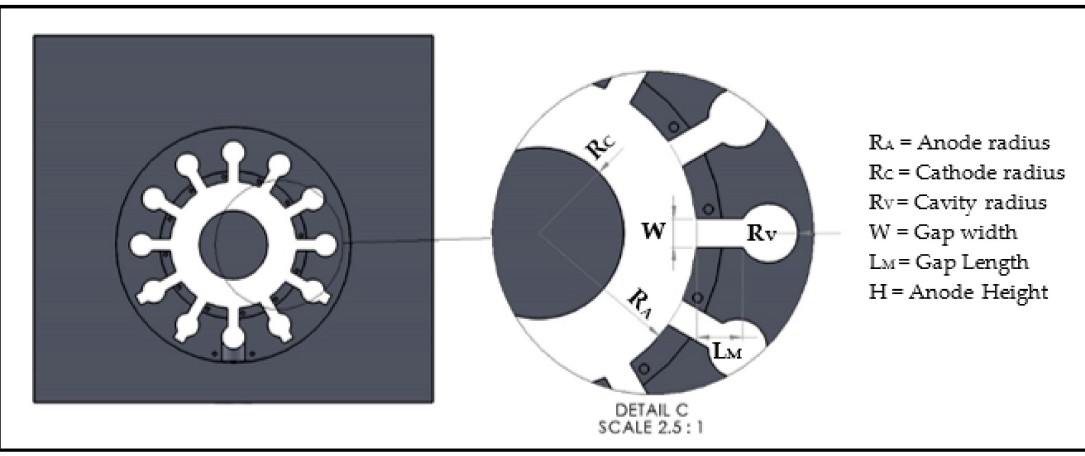

**Figure 6.** The section areas of the analytical equivalent resonant circuit system.

### 2.2.2. The Analysis of Resonant Frequency by PIC Simulation Method

Generally, CST Particle Studio (CST Studio Suite, License No. 5065f3510dac, Suranaree University of Technology) is used to calculate and perform particle dynamics in the 3D electric field with high accuracy. This commercial code consists of several solvers, such as the eigenmode solver, particle tracking and particle-in-cell (PIC). Each solver has different features for simulation. The tracking particle provides the charged particle trajectory in electrostatic, magnetostatic or eigenmode fields. The wake field is used to compute the fields generated from the interaction between the charged particles traveling with relativistic velocity and structure around them. The PIC solver deals with simulation of the charged particle in the time domain and performs consistent simulation of particles and electromagnetic fields. Using these features of CST, many devices, particularly high-power magnetrons, have been designed and simulated based on this code. In addition, it is possible to import all geometric parameters of a structural magnetron via Solid Works (Solid Works 2019, License No. 9020009109154195PTMG9JBC, Synchrotron Light Research Institute, Thailand) into the code and define the device materials needed for the simulation. Thus, the twelve-hole-slot-type magnetron is modeled and simulated using the particle-in-cell (PIC) method in CST. The normal procedures of the particle-in-cell (PIC) method are used for the interaction space between the electron motion in the electromagnetic field of the magnetron. The basic procedure behind the electromagnetic PIC solver is very similar to that demonstrated in the tracking simulation workflow. On the other hand, the tracking solver simulates particles in a self-consistent field by using an integration scheme for particles and electromagnetic fields, in line with the fundamentals of the PIC method algorithm, as shown in Figure 7 [22].

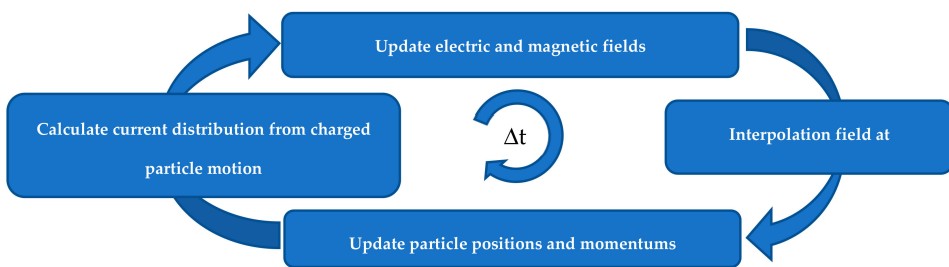

**Figure 7.** A block diagram of particle-in-cell (PIC) simulation technique.

The magnetron operation is simulated by the PIC simulation method. The magnetron consists of four components, namely the anode block with a hole-slot type and tuner slot distance component, cathode emission, double-strapped ring and vacuum components. These components were illustrated in 3D coordinates using Solid Works® and examples are shown in Figure 8.

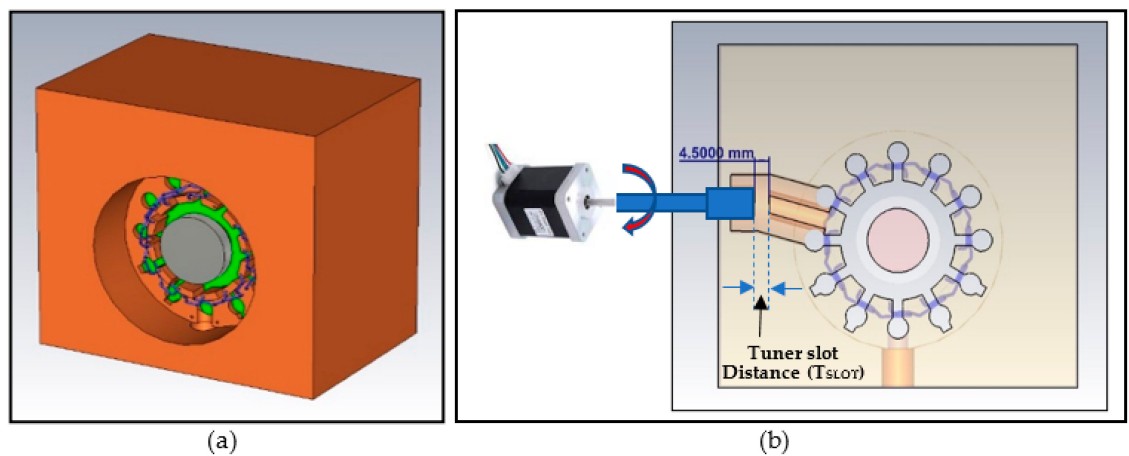

**Figure 8.** The twelve-hole-slot-type magnetron modeling. (**a**) 3D visual model; (**b**) tuning system of magnetron.

In Table 1, the parameters of the structural dimensions measured by the CMM are used to simulate magnetron operation in dominant mode. The resonant frequency is tuned by adjusting the tuner slot distance via the actuator. The tuner slot can mechanically move in and out with a range of 5 mm, which is done by a stepper motor. In the simulation, the particle distribution is uniform, based on an explosion emission model with an initial kinetic energy of 3 eV and particle density from 1830 emission points. In the simulation, conditions based on PIC are 45 kV DC voltage and a constant magnetic field density of 0.152 tesla. An anode voltage is applied for 300-ns duration with a rise time of 1 ns at 3 eV energy [23,24]. In addition to the analysis of the anode configuration, it can be defined by the properties of the materials, namely the electrical conductivity and electrical resistivity. Both electrical properties of the material under the ambient temperature of 20 Celsius can be analyzed by using the linear equation if the temperature is slightly changed. Thus, the equation of the electrical conductivity and resistivity can be expressed as follows [25].

$$\sigma = \sigma_{20}[1 + \alpha_{20}(\Delta T)], \tag{12}$$

where $\sigma = \frac{1}{\rho}$ is the electrical conductivity at a given temperature, $\sigma_{20}$ is the conductivity at ambient temperature 20 °C, $\alpha_{20}$ is the temperature coefficient at 20 °C and $\Delta T$ is the difference in temperatures between the temperature of the material and the fixed reference temperature of 20 °C.

**Table 1.** Specification of the cavity magnetron.

| Descriptions | Values |
|---|---|
| Number of resonant cavities | 12 |
| Cathode radius ($R_C$) | 9.160 mm |
| Anode radius ($R_A$) | 17.023 mm |
| Cavity radius ($R_V$) | 3.195 mm |
| Voltage applied ($V_{APP}$) | 45,000 v |
| Permanent magnet ($B_{APP}$) | 0.152 t |
| Tuner slot distance ($T_{SLOT}$) | 3.22–8.22 mm |

Using Equation (12), the key variables of electrical conductivity relating to ambient temperature can be analyzed through the coefficient of the resistivity or conductivity, which are slightly changed within the boundary and then multiply between the temperature difference and the temperature coefficient of the material to yield a value of less than 1. This affects the analysis of the estimation values by the linear equation or the simulation by the program that is adequate for the analysis of the material property.

### 2.3. Linear Acerelerator for the Sterilization System

Synchrotron Light Research Institute, Thailand has successfully installed and commissioned the LINAC 6 MeV sterilization application that is shown in Figure 9. The structure of the designed system is divided into several main sections. The electron gun is a diode thermionic gun, used to emit electrons with a peak current of 400 mA into the 6-MeV side-coupling LINAC. All emitted electrons are accelerated at an S-band resonant frequency of 2.9982 GHz by a longitudinal electric field generated from the 12-hole-slot magnetrons. At the LINAC exit, these electrons collide with a tungsten target, which emits photons in the form of X-rays; therefore, X-ray production is maintained at the desired dose for the entire irradiation process. All systems and sub-systems should work properly. For this LINAC, the electrons are accelerated up to 6 MeV with a resonant frequency of 2.9982 GHz at a temperature of 40 °C and a vacuum pressure of $1 \times 10^{-9}$ torr. One of the most crucial components for the reliable performance of X-ray production is an automatic frequency control (AFC) system. The electrons successfully accelerate only RF pulses from the magnetron and must be coupled with the LINAC cells. This means that the RF pulse must display exactly the correct resonant frequency. The resonant frequency of the magnetron can be

adjusted to correct the mismatch in the LINAC frequency as the machine operates by using the automatic frequency control (AFC) algorithm. The design of the AFC system depends on the RF signal taken from a directional coupler connected with a waveguide. It gives output signals, one of which displays forward power, and the other is the output signal obtained from the forward power and is referred to as the reflect power [26]. Both signals are compared by calculating the different phases. The difference in these signals is used as the error and the magnetron frequency is tuned to the resonant frequency. To reduce undesirable signals, high frequency and noise from the AC power supply are eliminated by using a low-pass filter. The difference in signal is used to generate the amplitude of the signal pulses with a proportional phase ratio of the frequency input—this is the principle of the AFC system's modules for designing the tuning frequency of the magnetron to match the accelerator tube continuously. Thus, the module of the AFC system is the key component for controlling the AFC system's reliability and stability, and the processes of the AFC system are shown in Figure 10. In addition to the tuning of the resonant frequency, the system is designed via the technique of an online adaptive parameter, referring to the condition that occurs as the Takagi–Sugeno model.

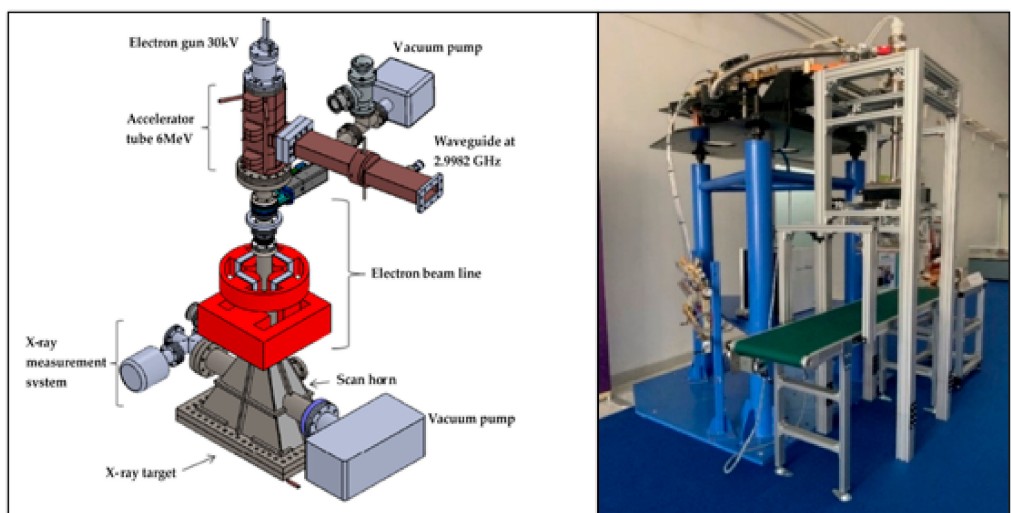

**Figure 9.** Linear Accelerator system for X-ray sterilization.

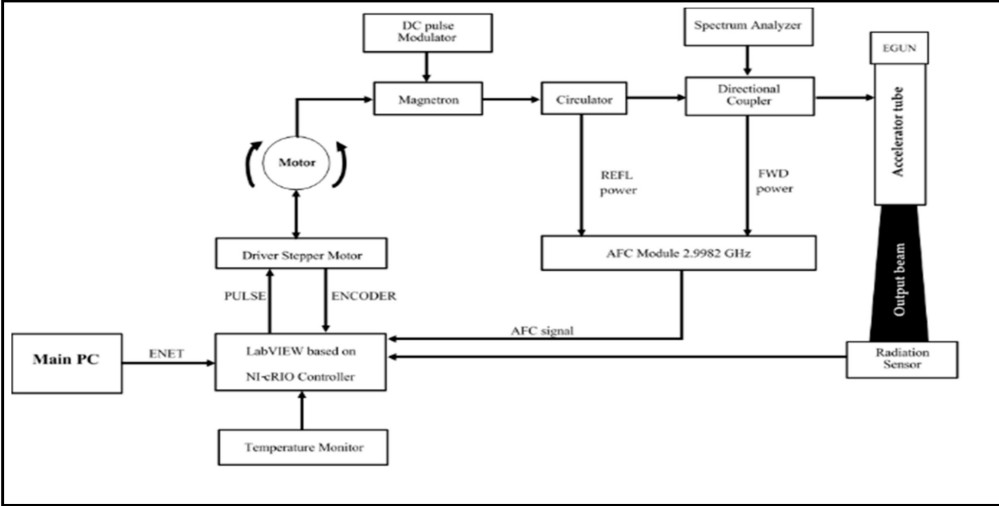

**Figure 10.** Block diagram of automatic frequency control.

### 2.3.1. Fuzzy Logic Control for the LINAC Sterilization

The design of the fuzzy logic control depends on the logic intelligent controller that uses the data, knowledge and experience of specialists without using the mathematical

model of the system. The FLC controller is appropriate for the complex, uncertain and multi-input system. The components of the FLC controller include preprocessing and postprocessing, adjusting the input and output data for controlling the suitable FLC system as required. The structures of the FLC control system are shown in Figure 11 and the designed rules based on the FLC depend on the knowledge base and experience of specialists. It then uses an inference engine in order to design the automatic frequency control of the LINAC sterilization system, as described by the block diagram given in Figure 12.

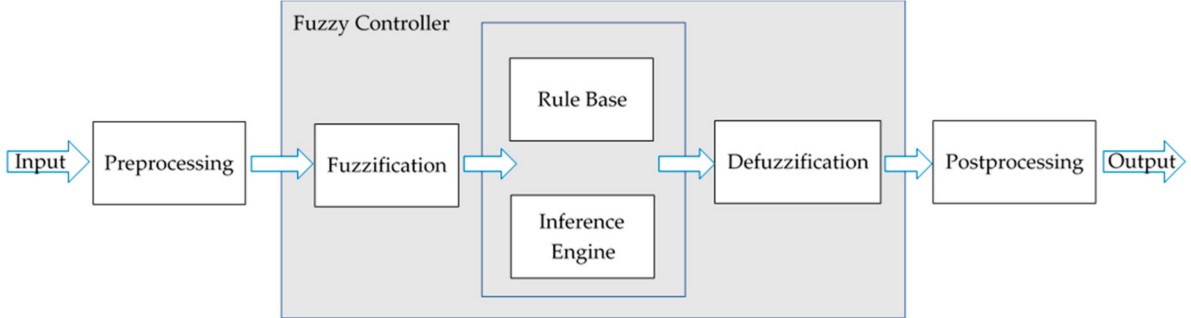

**Figure 11.** The structure of the fuzzy controller.

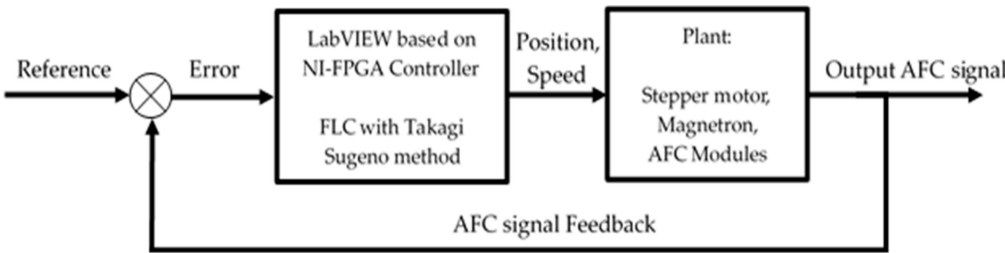

**Figure 12.** The block diagram of the automatic frequency control system.

In Figure 12, the error of the system is the difference in the feedback signal between the AFC-A and AFC-B compared with the reference input. The error data are calculated and applied in the FLC rule base. Before the data are applied to the fuzzy controller, the data must be converted from measured data to proper data based on the preprocessing function of the fuzzy system. The verified data of the FLC rule base with the inference engine are also converted by postprocessing in order to use suitable output data for the system as required. The position and speed data are used for controlling the stepping motor in order to adjust the structural dimensions of the resonance cavity of the magnetron. Thus, the results of the AFC signal are also changed.

### 2.3.2. Implementation of the Fuzzy Controller

The determination of the input in order to use the fuzzy rule base requires the input data's conversion or the process of fuzzification to obtain data for the suitable control system. This is described by the triangle-shaped membership functions graph that is shown in Figure 13a. Therefore, the converted data in the degree of membership can be expressed as in an Equations (13) and (14) and the results of the data of the input and output variables for the designed FLC are listed in Table 2.

$$\mu_x = \frac{X - A_0}{A_1 - A_0}; A_0 \leq X < A_1, \tag{13}$$

$$\mu_x = \frac{X - A_2}{A_1 - A_2}; A_1 \leq X \leq A_2, \tag{14}$$

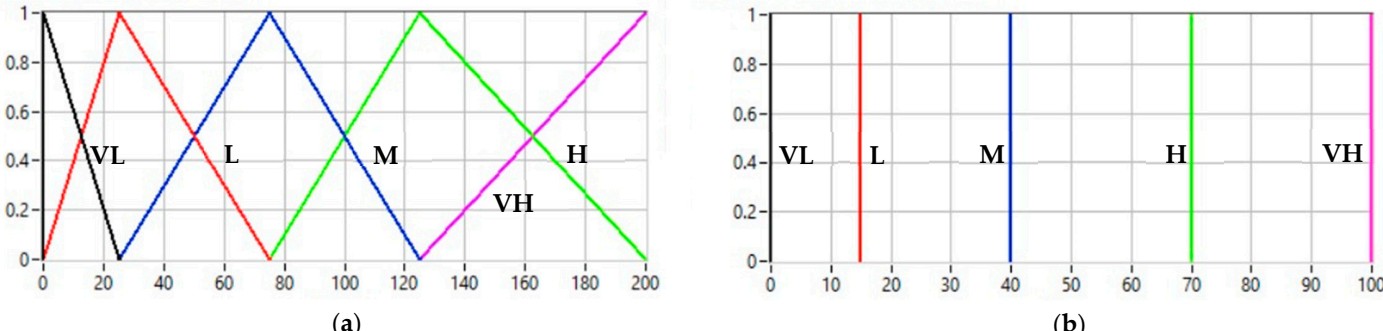

**Figure 13.** Membership function variables: (**a**) input variables; (**b**) output variables.

**Table 2.** Input and output variables for fuzzy logic control.

| No. | Input (Error) | Variable Input | Variable Output | Output (%) | Mean of Variable |
|-----|---------------|----------------|-----------------|------------|------------------|
| 1 | $\leq 0$ | VL | VL | 0 | Very Low |
| 2 | 25 | L | L | 15 | Low |
| 3 | 75 | M | M | 40 | Medium |
| 4 | 125 | H | H | 70 | High |
| 5 | $\geq 200$ | VH | VH | 100 | Very High |

For the desired output variable, it is converted to the suitable data for controlling the LINAC system or defuzzification process. This is the section in which the data of the FLC rule base are converted to the data of the real system by using a mathematical model, such as the methods of Mandani and Takagi–Sugeno. These two methods emphasize the specification, control function, model and method, but the difference in the mathematical model is the inference engine, for which the methods of Mandani and Takagi–Sugeno use the center of gravity (COG) and center of sums (COS), respectively. For this research, in the LINAC application, the automatic resonant frequency (AFC) for the LINAC system is applied using the Takagi–Sugeno method, which represents the inference engine of the membership function output variables developed by using the LabVIEW program (LabVIEW 2016, License No. M76X33883, Synchrotron Light Research Institute, Thailand). Thus, the output variables are defined by the singleton shape membership functions graph that is shown in Figure 13b, and the inference engine is designed using the Takagi–Sugeno method that is expressed in Equation (15) [27–29].

$$COS = \frac{\sum_{m=1}^{L} \mu_{(k_m)} k_m}{\sum_{m=1}^{L} \mu_{(k_m)}}, \tag{15}$$

where $\mu_{(k_m)}$ is the degree of input variable membership, $k_m$ is the output variable range.

## 3. Experimental Results

The experimental results of the LINAC sterilization system with 6 MeV are used in the reverse engineering method to identify the suitable parameters of the magnetron operation in dominant mode. The efficiency of controlling the resonant frequency of the twelve-hole-slot-type magnetron is evaluated in the research in four sections as follows.

### 3.1. The Results of the Operating Point of the Twelve-Hole-Slot-Type Magnetron

The operating point of the magnetron MG7095 model which is twelve-hole-slot-type depends on a suitable DC voltage and the application of a permanent magnetic field to the magnetron. This results in stability and continuous magnetron operation. These parameters are analyzed through the theoretical electron motion in 3D cylindrical coordinates and the

structure dimensions that are measured by CMM with tolerancing at 0.5 μm. The results of the dimensions of the geometry of the magnetron are given in Table 3, and the results of the Hull cut-off and Hartree condition with the resonant frequency as required at 2.9982 GHz as defined by applying an anode voltage of 45,000 volts and a permanent magnetic field of 0.152 tesla are shown in Figure 14.

**Table 3.** Structural dimensions of twelve-hole-slot-type magnetron.

| Component | Data (Millimeters) |
|---|---|
| Anode radius ($R_A$) | $17.023 \pm 0.00025$ |
| Cathode radius ($R_C$) | $9.160 \pm 0.00025$ |
| Cavity radius ($R_V$) | $3.195 \pm 0.00025$ |
| Height of anode (H) | $32.801 \pm 0.00025$ |
| Gap length ($L_M$) | $4.957 \pm 0.00025$ |

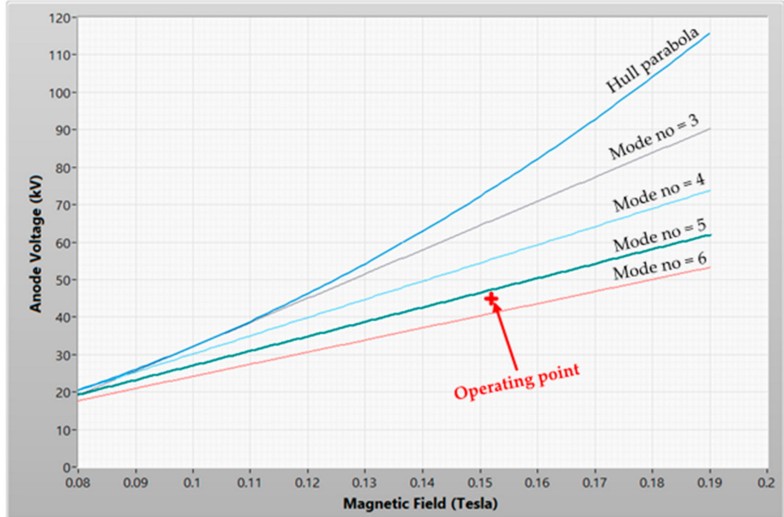

**Figure 14.** The results of calculation of a suitable voltage for the operating magnetron.

### 3.2. The Results of the Resonant Frequency by Using the Equivalent Parallel Circuit

The magnetron is found to achieve the dominant mode in the frequency domain by using the equivalent parallel circuit in the one-section network of the magnetron. The structural dimensions and the presence of the electric and magnetic fields in the magnetron are also considered and used to analyze how this mode is achieved. The results of the resonant frequency indicate the frequency of each mode of operation of the magnetron between modes 1 to 11 and then sets the $T_{SLOT}$ at 3.97 mm in the dominant mode or pi-mode operation (mode number = 6) as 2.99816 GHz. Compared with the target frequency of 2.9982 GHz, this equates to a difference of 0.04 MHz or a relative error of 0.0013%, and the mode of operation results for the magnetron are shown in Figure 15.

### 3.3. The Results of the Resonant Frequency by Using PIC Simulation Method

The simulation results are analyzed by using the PIC simulation method with the initial conditions defined in Table 4.

The PIC simulation was run based on the PC (Intel Core-i9, CPU3.1 GHz, and 32 GB-RAM) and the initial conditions listed in Table 4. The entire process of simulation takes 18 h and the simulation results of the electrical conductivity of annealed copper are ex-prssed in Equation (12), which substitutes the target temperature of the material with the temperature coefficient and the ambient temperature of 20 °C. The PIC simulation results at a temperature range of between 0 and 70 °C are shown in Figures 16 and 17.

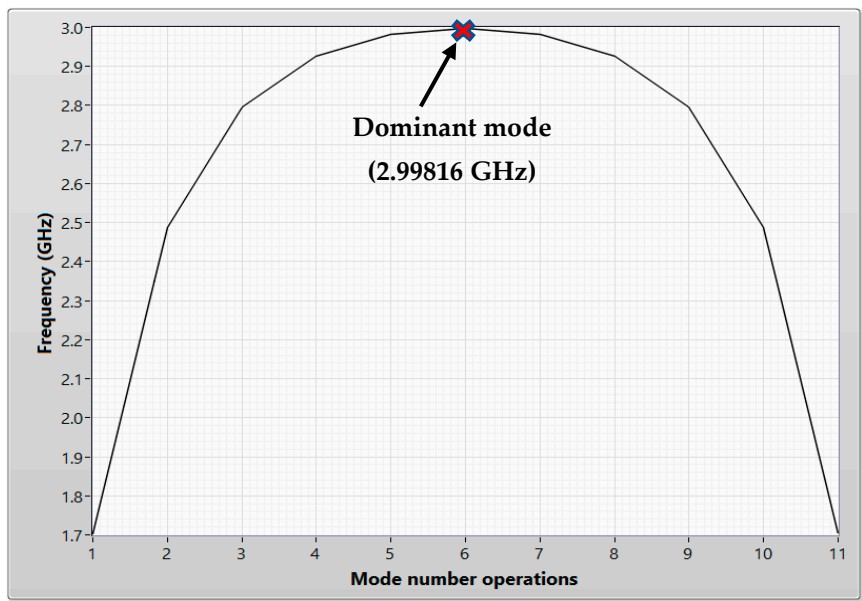

**Figure 15.** The result of resonant frequency for each mode of magnetron operation.

**Table 4.** The initial conditions of the simulation by Particle-In-Cell.

| Component | Data (Millimeters) |
|---|---|
| Kinetic energy | 3 eV |
| Rise time | 1 ns |
| Duration time | 150 ns |
| Anode material | Copper annealed |
| Electrical conductivity | $5.8 \times 10^7$ S/m |
| Temperature coefficient | $0.00393$ K$^{-1}$ |
| Ambient temperatures | 20 °C |

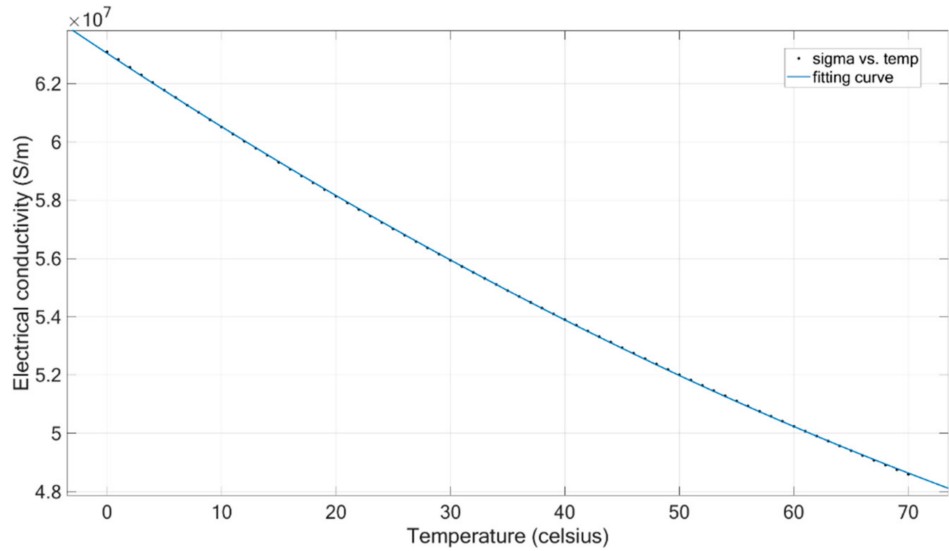

**Figure 16.** The electrical conductivity response after temperature adjustment.

The tolerance of the radial resonant cavity radius of the magnetron with machine tolerancing was 50 μm. The structural dimensions using the 3D coordinates were modeled in Solid Works®, resulting in defining the radius of the resonant cavity as 3.145 to 3.245 mm and the distance of the tuner slot ($T_{SLOT}$) as 3.22 to 8.22 mm. The position results of the resonant frequency obtained using the PIC simulation method indicate the range of the

capable tuner to be 11.7 MHz, approximately. The suitable tolerances of the resonant cavity are defined between the radius of 3.187 and 3.215 mm with a normal cavity radius as measured by CMM as 3.195 mm. Thus, the appropriate tolerances of the designed device for the resonant frequency of 2.9982 GHz are −8 μm to +20 μm. The results of each condition of the resonant cavity are expressed in the table below (Tables 5 and 6).

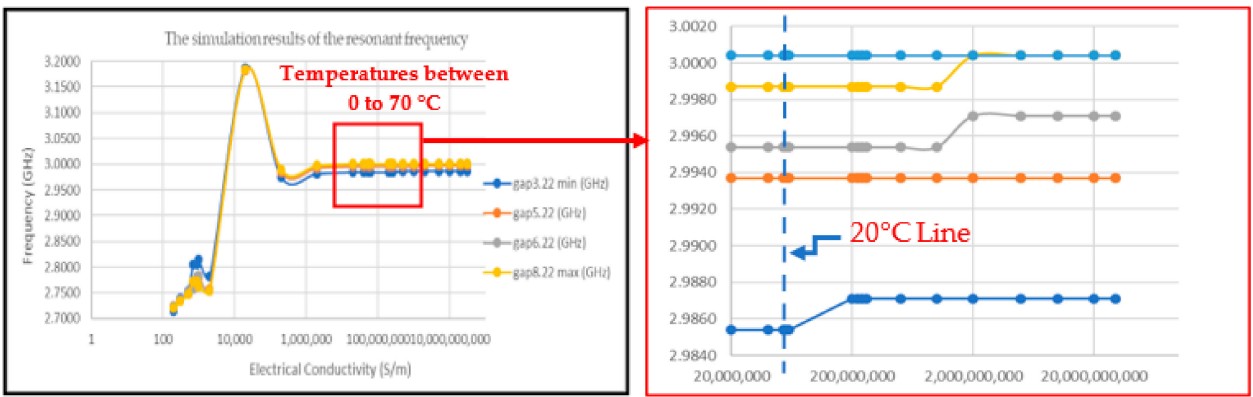

**Figure 17.** The simulation results of the resonant frequency with the adjustment of the electrical conductivity.

**Table 5.** Simulation results of the resonant frequency by PIC simulation method.

| Tuner Slot Distance (mm) | The Frequency of the Cavity Radius (GHz) | | | | | | | | | |
|---|---|---|---|---|---|---|---|---|---|---|
| | R3.145 | R3.170 | R3.185 | R3.187 | R3.190 | R3.195 | R3.200 | R3.207 | R3.215 | R3.220 | R3.245 |
| 2.22 | 3.0083 | **2.9983** | 2.9933 | 2.9900 | 2.9883 | 2.9867 | 2.9850 | 2.9833 | 2.9800 | 2.9783 | 2.9650 |
| 2.97 | 3.0150 | 3.0033 | **2.9983** | 2.9950 | 2.9950 | 2.9933 | 2.9917 | 2.9883 | 2.9867 | 2.9850 | 2.9700 |
| 3.22 | 3.0167 | 3.0050 | 3.0000 | 2.9967 | 2.9967 | 2.9950 | 2.9933 | 2.9900 | 2.9883 | 2.9867 | 2.9717 |
| 3.47 | 3.0167 | 3.0067 | 3.0017 | **2.9983** | 2.9967 | 2.9967 | 2.9950 | 2.9917 | 2.9900 | 2.9867 | 2.9733 |
| 3.72 | 3.0183 | 3.0083 | 3.0033 | 3.0000 | **2.9983** | 2.9967 | 2.9950 | 2.9933 | 2.9917 | 2.9883 | 2.9750 |
| 3.97 | 3.0200 | 3.0083 | 3.0033 | 3.0000 | 3.0000 | **2.9983** | 2.9967 | 2.9933 | 2.9917 | 2.9900 | 2.9767 |
| 4.22 | 3.0217 | 3.0117 | 3.0067 | 3.0033 | 3.0017 | 3.0000 | **2.9983** | 2.9967 | 2.9933 | 2.9917 | 2.9783 |
| 5.22 | 3.0233 | 3.0133 | 3.0083 | 3.0050 | 3.0050 | 3.0033 | 3.0017 | **2.9983** | 2.9967 | 2.9950 | 2.9800 |
| 6.22 | 3.0267 | 3.0167 | 3.0117 | 3.0083 | 3.0067 | 3.0050 | 3.0033 | 3.0017 | **2.9983** | 2.9967 | 2.9833 |
| 7.22 | 3.0267 | 3.0167 | 3.0117 | 3.0083 | 3.0067 | 3.0067 | 3.0033 | 3.0017 | 3.0000 | 2.9967 | 2.9833 |
| 8.22 | 3.0283 | 3.0167 | 3.0117 | 3.0100 | 3.0083 | 3.0067 | 3.0050 | 3.0017 | 3.0000 | **2.9983** | 2.9850 |
| 9.22 | 3.0283 | 3.0167 | 3.0117 | 3.0100 | 3.0083 | 3.0067 | 3.0050 | 3.0033 | 3.0000 | **2.9983** | 2.9850 |

**Table 6.** Simulation results of the resonant frequency by the equivalent circuit method.

| Tuner Slot Distance (mm) | The Frequency of the Cavity Radius (GHz) | | | | | | | | | |
|---|---|---|---|---|---|---|---|---|---|---|---|
| | R3.145 | R3.170 | R3.185 | R3.187 | R3.190 | R3.195 | R3.200 | R3.207 | R3.215 | R3.220 | R3.245 |
| 2.22 | 3.00855 | 2.99734 | 2.99068 | 2.98979 | 2.98847 | 2.98626 | 2.98406 | 2.98098 | 2.97748 | 2.97530 | 2.96446 |
| 2.97 | 3.01558 | 3.00435 | 2.99767 | 2.99678 | 2.99545 | 2.99324 | 2.99103 | 2.98795 | 2.98444 | 2.98226 | 2.97139 |
| 3.22 | 3.01733 | 3.00610 | 2.99942 | 2.99853 | 2.99720 | 2.99499 | 2.99278 | 2.98970 | 2.98619 | 2.98400 | 2.97313 |
| 3.47 | 3.01798 | 3.00674 | 3.00006 | 2.99917 | 2.99784 | 2.99563 | 2.99342 | 2.99034 | 2.98682 | 2.98464 | 2.97376 |
| 3.72 | 3.01925 | 3.00800 | 3.00132 | 3.00043 | 2.99910 | 2.99689 | 2.99468 | 2.99159 | 2.98808 | 2.98589 | 2.97502 |
| 3.97 | 3.02138 | 3.01013 | 3.00344 | 3.00255 | 3.00122 | 2.99901 | 2.99680 | 2.99371 | 2.99020 | 2.98800 | 2.97712 |
| 4.22 | 3.02388 | 3.01263 | 3.00593 | 3.00504 | 3.00371 | 3.00149 | 2.99928 | 2.99619 | 2.99268 | 2.99048 | 2.97959 |
| 5.22 | 3.02609 | 3.01483 | 3.00813 | 3.00724 | 3.00590 | 3.00369 | 3.00147 | 2.99838 | 2.99486 | 2.99267 | 2.98177 |
| 6.22 | 3.02755 | 3.01628 | 3.00958 | 3.00869 | 3.00735 | 3.00513 | 3.00292 | 2.99983 | 2.99631 | 2.99411 | 2.98321 |
| 7.22 | 3.02821 | 3.01693 | 3.01023 | 3.00934 | 3.00801 | 3.00579 | 3.00357 | 3.00048 | 2.99696 | 2.99476 | 2.98386 |
| 8.22 | 3.02844 | 3.01717 | 3.01047 | 3.00958 | 3.00824 | 3.00602 | 3.00381 | 3.00072 | 2.99719 | 2.99500 | 2.98409 |
| 9.22 | 3.02886 | 3.01759 | 3.01088 | 3.00999 | 3.00866 | 3.00644 | 3.00422 | 3.00113 | 2.99761 | 2.99541 | 2.98451 |

In Figure 17, the simulation results with the adjusted electrical conductivity obtained a resonant frequency that was not changed and can be separated from the variable of

electrical conductivity. Next, the simulation results of the adjusted T$_{SLOT}$ in the range of 3.22 to 8.22 mm indicate an optimal value of 11.7 MHz, and examples of each tuner slot distance are shown in Figures 18–20.

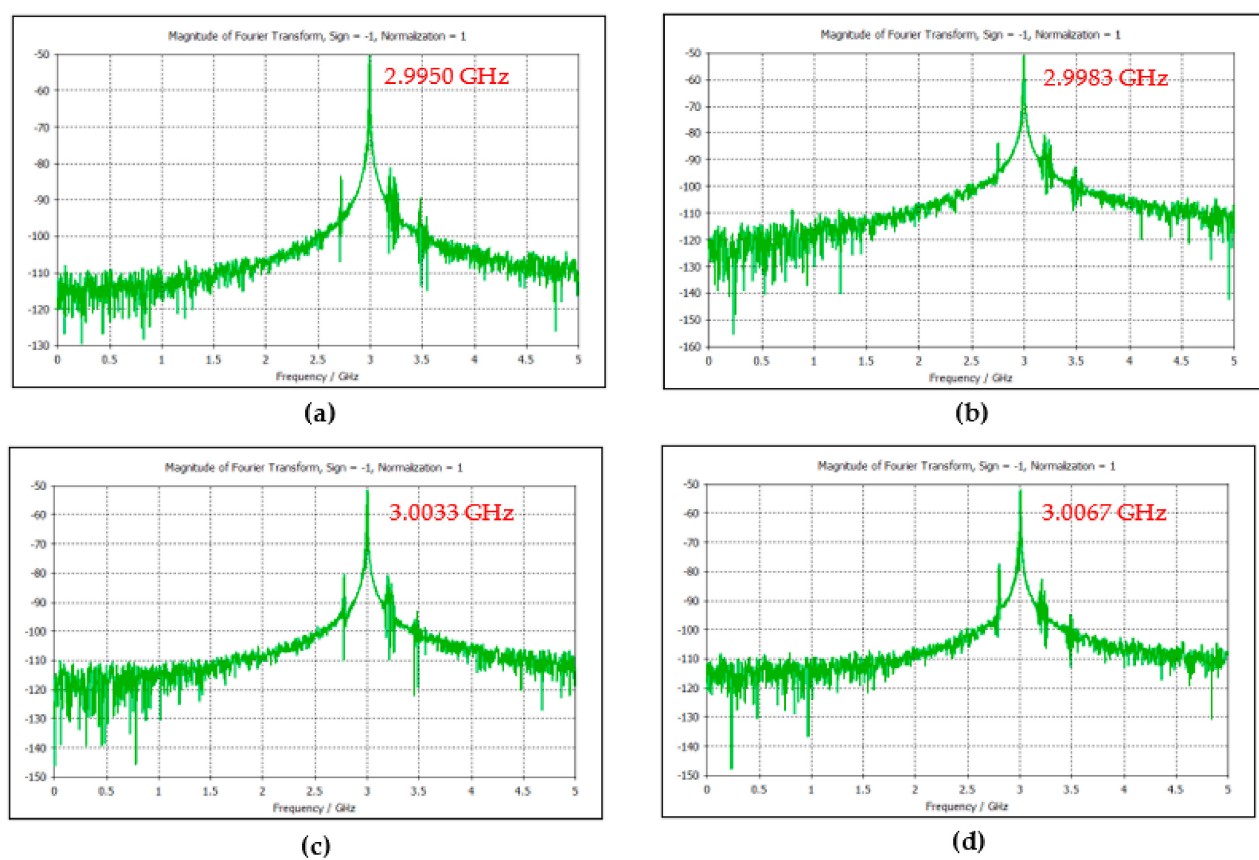

**Figure 18.** The results of the resonant frequency position for each tuner slot distance: (**a**) 3.22 mm; (**b**) 3.97 mm; (**c**) 5.22 mm; (**d**) 8.22 mm.

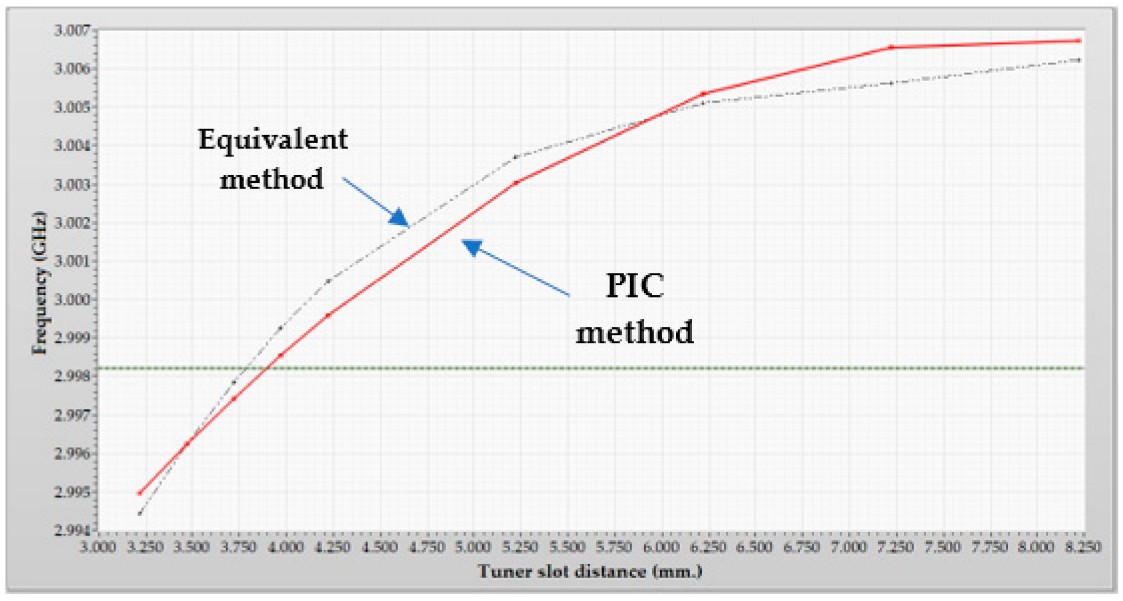

**Figure 19.** The simulation results of resonant frequency after adjusting tuner slot distance.

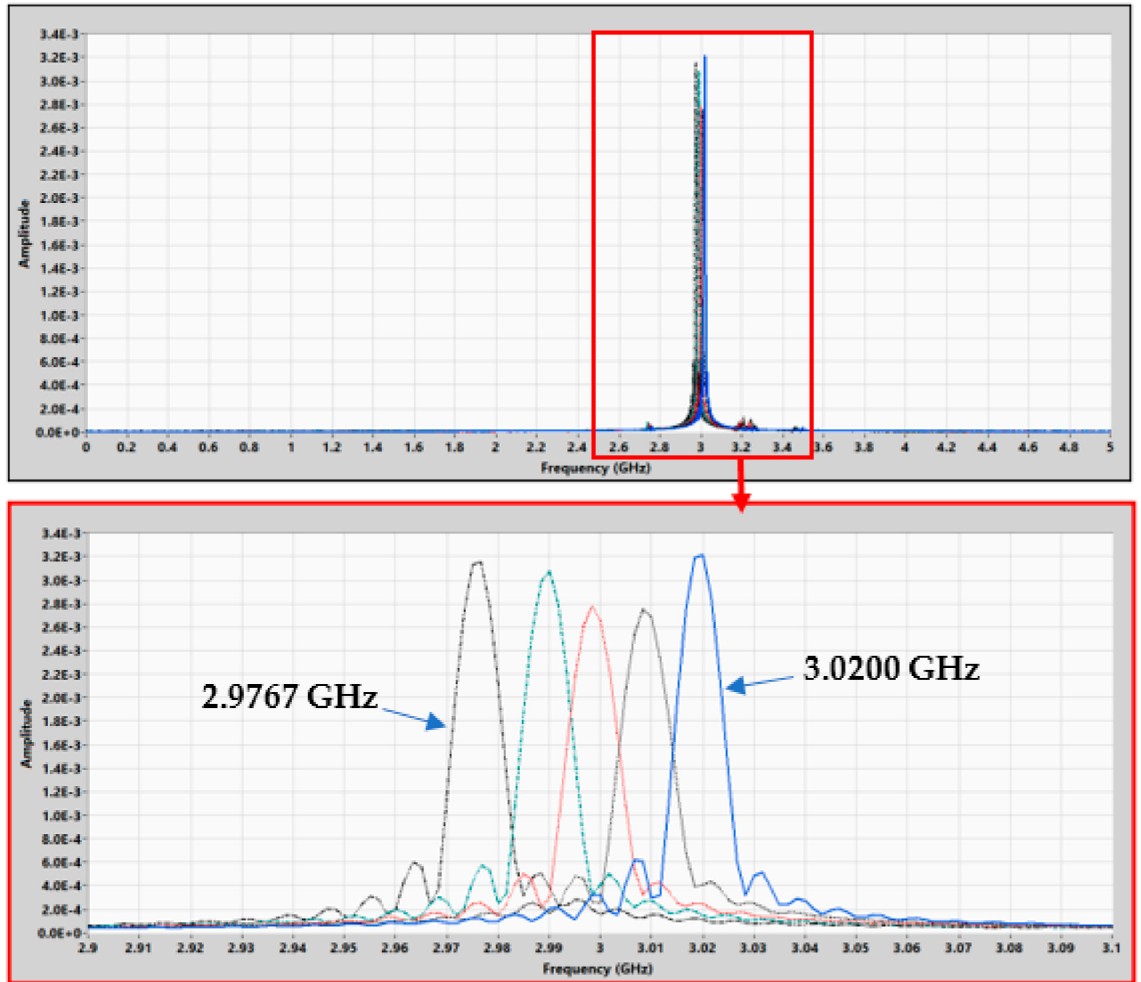

**Figure 20.** The spectrum of the resonant frequency of each cavity radius with tuner distance of 3.97 mm.

### 3.4. Results of Experimental Setup

The LINAC sterilization application is controlled by the FLC algorithm and Takagi–Sugeno method. The experimentation sets a cooling temperature of 40 °C and the vacuum pressure of the LINAC is less than $5 \times 10^{-8}$ torr. The suitable parameters are measured by CMM and calculated using the equation of the Hull cut-off and Hartree conditions, determining an anode voltage and permanent magnetic field of 45 kV and 0.152 tesla, respectively. The X-ray responses of the LINAC are shown in Figure 19.

In Figure 21, the results of the designed FLC controller under a duration time of 45 min indicate that the response of the X-ray level is constant at 0.8 μSV/h within 12 min (see dot-rectangular #1) and 20 min (see dot-rectangular #1), respectively. During the experimentation, the X-ray dose rates decreased suddenly (see dot-circle #1 and #2) because the temperature changed by 1.5 Celsius, as illustrated in Figure 21d. The result of the resonant frequency generated by the magnetron shifted beyond the target frequency of 2.9982 GHz, and the frequency response converges at the desired boundary of the frequency if the system is controlled by the FLC, which compensates for the frequency difference by adjusting the tuner slot distance via a stepping motor, as shown in Figure 21c. This result of the X-ray response is constant. The compensation of the frequency depends on the AFC signal feedback or the difference in the forward and reflect signals compared with the reference frequency at 0 volts or 2.9982 GHz, and the motor position response is shown in Figure 21b.

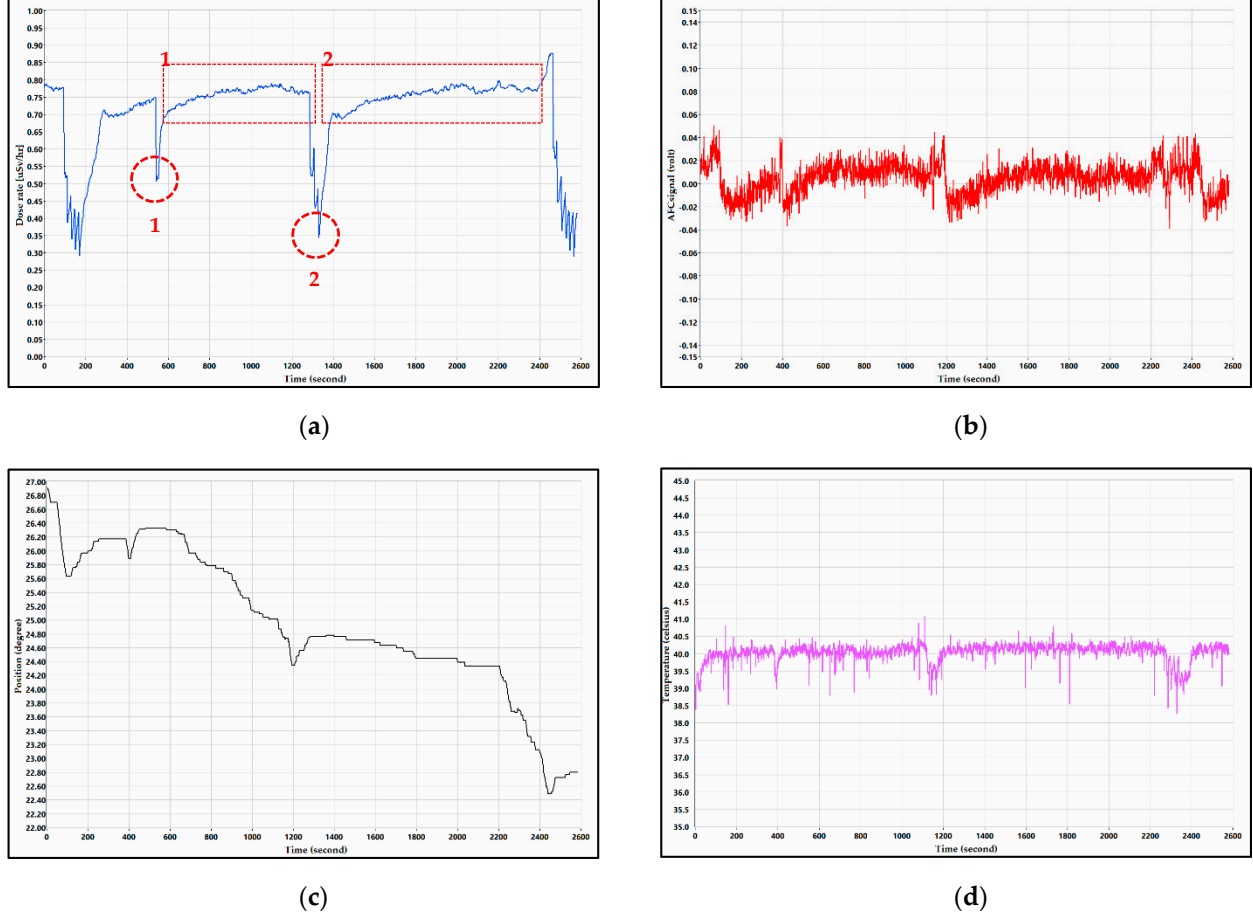

**Figure 21.** The fuzzy system response: (**a**) X-ray response; (**b**) AFC signal response; (**c**) Motor position response; (**d**) Temperature response.

## 4. Conclusions

This paper presents the analysis results of a twelve-hole-slot-type magnetron designed to fulfil the suitable parameters based on dominant-mode (pi-mode) operation. In the experiments, in both relativistic and theoretical magnetrons, all parameters and boundaries of the suitable resonant frequency could be determined, which are usually used for the design and manufacturing of a magnetron prototype to reach a desired tolerance in the radius of the cavity. The following conclusions about the estimated resonant frequency can be drawn from this work:

1.  The estimation of the operating point of the magnetron in dominant mode was conducted, generating a resonant frequency of 2.9982 GHz, after measuring the geometry of the inner structure of the magnetron using a CMM with a resolution of 0.5 μm. The results of the normal cavity radius yielded a tolerance value of $3.195 \pm 0.00025$ mm. The results of the boundary condition to operate the magnetron were an anode voltage range between 41.0 and 74.2 kV and a permanent magnetic field of around 0.152 tesla.

2.  The estimation of the resonant frequency was done by using the equivalent parallel circuit of the magnetron in operation modes 1 to 11. This resulted in a resonant frequency in pi-mode at the desired anode voltage of 45 kV with $T_{SLOT}$ position of 3.97 mm. The measured cavity radius of $3.195 \pm 0.00025$ mm has a relative error of 0.02% with reference to the target frequency load of 2.9982 GHz.

3.  The resonant frequency can be estimated by using PIC simulation in the CST Particle Studio program. The results of the resonant frequency in pi-mode with $T_{SLOT}$ position

as 3.97 mm and a cavity radius of 3.195 $\pm$ 0.00025 mm yield a relative error of 0.03% with reference to the target frequency of the accelerator tube.

4. The design and the manufacturing of the prototype of the magnetron with tolerance are required for PIC simulation. Based on the standard measured cavity radius of 3.195 $\pm$ 0.00025 mm, a cavity radius in the range of $-8$ μm to $+20$ μm and $T_{SLOT}$ tuning of 3.22 to 8.22 mm are sufficient to tune the resonant frequency of the magnetron equivalent to a LINAC frequency of 2.9982 GHz. Thus, the manufacturing of the prototype of the magnetron at a resonant frequency of 2.9982 GHz is required by a CNC machine which has a tolerance of less than $\pm14$ μm, approximately. The advantages of a reverse engineering technique are studied to reach the structural dimensions and solve the operating point of the magnetron in dominant mode.

5. The control system for magnetron is based on the fuzzy logic of the Takagi–Sugeno method agrees well with the LINAC's frequency with an anode voltage supply in pi-mode of 45-kV and magnetic field of permanent magnet of 0.15 T.

**Author Contributions:** Conceptualization, N.Y., S.C. and J.S.; methodology and validation, N.Y., J.S., S.C. and N.R.; software and formal analysis, N.Y., S.C., N.R. and J.S.; investigation, N.Y. and J.S.; writing—original draft preparation, N.Y.; writing—review and editing, N.Y., J.S., S.C. and N.R. All authors have read and agreed to the published version of the manuscript.

**Funding:** This research was funded by the Ministry of Higher Education Science Research and Innovation, grant number 5401/1324.

**Institutional Review Board Statement:** Not applicable.

**Informed Consent Statement:** Not applicable.

**Data Availability Statement:** Not applicable.

**Acknowledgments:** The authors would like to thank the Ministry of Higher Education Science Research and Innovation, Synchrotron Light Research Institute, and Suranaree University of Technology for granting budgets, equipment, and locations.

**Conflicts of Interest:** The authors declare no conflict of interest.

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
