# Peer review of "Parameter Optimization of Hole-Slot-Type Magnetron for Controlling Resonant Frequency of Linear Accelerator 6 MeV by Reverse Engineering Technique"

_applsci, doi:10.3390/app11052384_

Round 1
Reviewer 1 Report
This paper presents parameters optimization of a twelve-hole-slot type magnetron based on reverse engineering technique. Its focus is on the effects of disturbances such temperature and vacuum pressure.
Dear authors, the work has novelty and is of interest to readers. However, you are using figures (figs. 10, 12, 13, 14,) that the reader may see in your earlier work: N. Yachum, N. Russamee and J. Srisertpol, Automatic frequency control of the magnetron system for medical linear accelerator using fuzzy logic control, Proceedings of XLVI International Summer School-Conference APM 2018, pp. 275-285. I recommend you to reflect the results presented in the indicated figures in a different way. I believe that the article will not lose its high quality. Wish you luck
I believe that the article can be published after minor revision
Author Response
Dear Reviewer
I sent my revised manuscript with response to your comments. Please see attached file as your request.
Thank you for your consideration of this manuscript.
Sincerely
Srisertpol J, PhD.

Reviewer 2 Report
- There is a typo on page 3: “The structure of the paper is given as follow; In section 2, describes the research methodology that..”.
- Also on page 3 is a typo: “development of the medical linear accelerator for the treatment cancer applications. [28] The”.
- There is a typo on page 5: “In the Figure 2, The method of measuring structure dimensions of the magnetron, using probe tip..”.
- Equation (4) contains an irrational entry: the last two multipliers on the right side can be combined and written more compactly. Also from the following it appears that n is not necessarily equal to N/2?
- Equation (10) contains a misprint: can't be a summand RV/12 for reasons of proper dimensionality.
- After Equation (10) is no decryption what is ΔT.
- Equation (15) contains perhaps a typo: the sum sign should be in both the numerator and the denominator?
- In “References” is no dot at the end after most links.
Author Response

(The authors gave the same response as above.)

Reviewer 3 Report
1365 / 5000Výsledky pÅ™ekladu
The authors of the article present the optimization of the parameters of a magnetron of the twelve-hole slit type based on the technique of reverse engineering in order to improve the operation of the 6 MeV Linear accelerator (LINAC) for fruit sterilization. Based on analyzes and experimental verifications, they show the results of the optimal design. I recommend to publishAuthor Response
Dear Reviewer
Thank you for your kind support, I sent my revised manuscript.
Thank you for your consideration of this manuscript.
Sincerely
Srisertpol J, PhD.